# Plasma Tsukushi Concentration Is Associated with High Levels of Insulin and FGF21 and Low Level of Total Cholesterol in a General Population without Medication

**DOI:** 10.3390/metabo12030237

**Published:** 2022-03-10

**Authors:** Masato Furuhashi, Yukimura Higashiura, Akiko Sakai, Masayuki Koyama, Marenao Tanaka, Shigeyuki Saitoh, Kazuaki Shimamoto, Hirofumi Ohnishi

**Affiliations:** 1Department of Cardiovascular, Renal and Metabolic Medicine, Sapporo Medical University School of Medicine, Sapporo 060-8543, Japan; corosukecorocororin8@icloud.com (Y.H.); akiko1119jp@yahoo.co.jp (A.S.); masa3yuki3@hotmail.com (M.K.); tanakamarenao@yahoo.co.jp (M.T.); ssaitoh@sapmed.ac.jp (S.S.); hohnishi@sapmed.ac.jp (H.O.); 2Department of Public Health, Sapporo Medical University School of Medicine, Sapporo 060-8543, Japan; 3Department of Nursing, Division of Medical and Behavioral Subjects, Sapporo Medical University School of Health Sciences, Sapporo 060-8556, Japan; 4Japan Health Care College, Sapporo 004-0839, Japan; k_shimamoto@jhu.ac.jp

**Keywords:** Tsukushi, hepatokine, nonalcoholic fatty liver disease, fatty liver index, insulin resistance, cholesterol, fibroblast growth factor 21, adiponectin, endoplasmic reticulum stress, inflammation

## Abstract

Tsukushi (TSK) is a member of the small leucine-rich proteoglycan family that controls developmental processes and organogenesis. TSK was also identified as a new hepatokine, which is mainly expressed in the liver, and is secreted by hepatocytes, to regulate energy and glycolipid metabolism in response to nonalcoholic fatty liver disease. However, the role of plasma TSK, especially its role in the general population, has not been fully addressed. We investigated the associations between plasma TSK concentration and several metabolic markers, including fibroblast growth factor 21 (FGF21), a hepatokine, and adiponectin, an adipokine, in 253 subjects (men/women: 114/139) with no medication in the Tanno–Sobetsu Study, which employed a population-based cohort. There was no significant sex difference in plasma TSK concentration, and the level was positively correlated with the fatty liver index (FLI) (r = 0.131, *p* = 0.038), levels of insulin (r = 0.295, *p* < 0.001) and levels of FGF21 (r = 0.290, *p* < 0.001), and was negatively correlated with the total cholesterol level (r = −0.124, *p* = 0.049). There was no significant correlation between the TSK level and body mass index, waist circumference, adiponectin, high-density lipoprotein cholesterol or total bile acids. The multivariable regression analysis showed that high levels of insulin and FGF21 and a low level of total cholesterol were independent determinants of plasma TSK concentration, after adjustment for age, sex and FLI. In conclusion, plasma TSK concentration is independently associated with high levels of insulin and FGF21, a hepatokine, and a low level of total cholesterol, but not with adiposity and adiponectin, in a general population of subjects who have not taken any medications.

## 1. Introduction

Tsukushi (TSK) is a member of the small leucine-rich proteoglycan family [1]. TSK was originally identified in the lens of chicks, and this molecule was named after its expression pattern in chick embryos, which is similar to the shape of the Japanese horsetail plant, Tsukushi [2]. TSK has been shown to be associated with various physiological processes, including growth and development, wound healing, and cartilage formation [3]. Studies in chicks and Xenopus showed that TSK controls different signaling pathways, including bone morphogenetic protein, β-catenin/Wnt, mitogen-activated protein kinase/fibroblast growth factor and transforming growth factor-β signaling pathways, during development and organogenesis, by interacting with various soluble factors and receptors [4].

It has recently been reported that TSK is induced in response to nonalcoholic fatty liver disease (NAFLD), and TSK has been identified as a hepatokine that is mainly expressed in the liver and is secreted by hepatocytes [5,6]. Both endoplasmic reticulum (ER) stress and inflammation have been reported to promote the expression and release of TSK in mice [5,6]. Gain- and loss-of-function studies in mice also showed that TSK has an impact on systemic cholesterol homeostasis [5]. TSK has been shown to lower the circulating high-density lipoprotein (HDL) cholesterol level, reduce cholesterol efflux capacity, and decrease cholesterol conversion to bile acids in the mouse liver [5]. Therefore, TSK could be a biomarker of liver stress that can link NAFLD to the development of atherogenic dyslipidemia and atherosclerosis.

It has recently been reported that the loss of TSK in mice increases sympathetic innervation and thermogenesis in brown adipose tissue, protects against diet-induced obesity and improves glucose homeostasis [7]. It has also been shown that the hepatic expression and plasma concentration of TSK are induced by feeding and are regulated by melanocortin-4 receptor signaling, and that TSK deficiency in mice augments feeding-induced c-Fos expression in the paraventricular nucleus of the hypothalamus, illustrating physiological crosstalk between TSK and the central regulatory circuit in maintaining energy balance and metabolic homeostasis [8]. However, another group showed that ablation of TSK did not affect the thermogenic capacity in brown adipose tissue, did not protect against the development of obesity and did not show improvements in glucose metabolism [9]. Overexpression of TSK in mice also failed to modulate thermogenesis, body weight and glucose tolerance [9]. Differences in genetic background, and variations in diet composition, housing conditions and gut microbiota composition may have affected the biological outcomes [7,9]. Therefore, the role of TSK in energy and glucose metabolism remains controversial.

Plasma TSK has been reported to be markedly elevated in mice with genetically induced and diet-induced obesity, as well as in mice with diet-induced nonalcoholic steatohepatitis [5,6,7]. It has also been shown that hepatic TSK expression is associated with steatosis in humans, and that the circulating level of TSK is markedly increased in patients suffering from acetaminophen-induced acute liver failure, a condition linked to severe hepatic inflammation [5]. However, the role of plasma TSK, especially its role in the general population, has not been fully addressed. It is also possible that TSK levels are modulated by several drugs, including anti-hypertensive and lipid-lowering drugs. Therefore, in the present study, we investigated the associations between plasma TSK concentration and several metabolic markers, including fibroblast growth factor 21 (FGF21), a hepatokine, and adiponectin, an adipokine, in a general population without medication, to avoid the effects of drugs on the TSK level.

## 2. Results

### 2.1. Characteristics of the Study Subjects

The basal characteristics of the 253 recruited subjects without medication (men/women: 114/139) are shown in Table 1. The numbers of subjects with smoking and alcohol drinking habits were 61 (24.1%) and 103 (40.7%), respectively. The numbers of male and female subjects with current smoking and alcohol drinking habits were 37/24 (32.5%/17.3%) and 61/42 (53.5%/30.2%), respectively, which are similar to the results obtained for individuals aged 50–60 years in the National Health and Nutrition Survey (https://www.nibiohn.go.jp/eiken/kenkounippon21/en/eiyouchousa/index.html, accessed on 9 March 2022) in Japan. Hypertension, diabetes mellitus and dyslipidemia were found in 66 (26.1%), 3 (1.2%), and 115 (45.5%) subjects, respectively. Men had significantly larger BMI and waist circumference measurements, significantly higher frequencies of habits of current smoking and alcohol drinking, and higher levels of blood pressure, liver enzymes, FLI, total bile acids, and parameters of renal function and glucose and lipid metabolism, except for cholesterols and insulin, than women.

The level of adiponectin was significantly lower in men than in women. There were trends of higher levels of FGF21 and TSK in men than in women, but there was no significant sex difference in the levels of FGF21 or TSK (Table 1). The plasma concentrations of TSK were comparable in subjects with and without a current smoking habit (mean [interquartile ranges]: 31 (16–46) vs. 29 (18–50) ng/mL, respectively; *p* = 0.390). There was also no significant difference between plasma TSK concentrations in subjects with and without an alcohol drinking habit (29 (17–46) vs. 29 (18–52) ng/mL, respectively; *p* = 0.351).

### 2.2. Correlations and Associations of Plasma Tsukushi with Parameters

As shown in Table 2, plasma TSK concentration was positively correlated with FLI (r = 0.131, *p* = 0.038) (Figure 1A), insulin (r = 0.295, *p* < 0.001) (Figure 1B), HOMA-R (r = 0.293, *p* < 0.001) and FGF21 levels (r = 0.290, *p* < 0.001) (Figure 1C), and was negatively correlated with total cholesterol level (r = −0.124, *p* = 0.049) (Figure 1D). When sex was separately analyzed, similar correlations of TSK level with insulin and HOMA-R were found (Table 2). Plasma TSK concentration was significantly correlated with levels of FLI, FGF21 and total cholesterol in women, but not in men. There were no significant correlations between TSK level and body mass index, waist circumference and levels of adiponectin, HDL cholesterol and total bile acids.

As a consideration of multicollinearity, insulin with a lower AIC score, but not HOMA-R with a higher AIC score, was incorporated in further multivariable regression analyses for plasma TSK concentration. As the best-fit model, high levels of insulin and FGF21 and a low level of total cholesterol were independent determinants of plasma TSK concentration, after adjustment for age, sex and FLI, explaining 15.4% of the variance (R^2^ = 0.154) (Table 3).

## 3. Discussion

The present study showed, for the first time, that a high plasma TSK concentration is independently associated with high levels of insulin and FGF21, a hepatokine, and a low level of total cholesterol in a general population of subjects without medication. Plasma TSK concentration was positively, but not independently, associated with FLI, and there was no significant correlation of TSK level with adiposity or adiponectin. TSK has been shown to be a new hepatokine, with circulating levels linked to hepatic fat accumulation, inflammation and ER stress in mice [5,6]. In humans, the expression level of TSK in the liver has been reported to be related to the degree of liver steatosis and liver injury [5,6]. The absence of an independent association between TSK and FLI in the present study is probably due to the lack of patients with severe NAFLD and liver damage. On the other hand, it has been reported that TSK is a negative regulator of adipose tissue sympathetic innervation, adrenergic signaling and thermogenesis in mice [7], and that there is physiological crosstalk between TSK and the central regulatory circuit in maintaining energy balance and metabolic homeostasis [8], though the role of TSK in energy metabolism remains controversial [9]. The present study suggests that TSK is mainly involved in hyperinsulinemia, as a marker of insulin resistance and the level of FGF21, a hepatic stress-induced hepatokine, rather than obesity and the level of adiponectin, an adipose-derived factor adipokine, in a general population.

It has been shown that hepatic TSK acutely modulates gene expression in the mouse liver [5], indicating that TSK acts, at least, via an autocrine and/or paracrine mechanism. TSK is an extracellular protein that can locally affect the activity of several signaling pathways in chicks and Xenopus [3,4]. TSK may directly bind to a still unidentified membrane receptor, or even interact with proteins in the plasma to modulate their functions. TSK might be directly involved in impaired insulin signaling, resulting in insulin resistance.

On the other hand, FGF21 is an endocrine molecule, mainly derived from the liver, that regulates glucose and lipid metabolism [10,11]. Treatment with FGF21 has been shown to improve glucose and lipid homeostasis, ameliorate hepatic steatosis [12], preserve β-cell functions [13], increase the number of brown adipocytes [14], and decrease atherosclerosis [15]. The concentration of FGF21 has been shown to be increased in several aspects of metabolic syndrome [16,17], indicating the presence of a compensatory response to higher metabolic stress or resistance to FGF21. It remains unclear whether FGF21 and TSK can regulate each other in connection with insulin resistance. In addition, it has been reported that obesity is characterized by reciprocal alterations in FGF19 (decrease) and FGF21 (increase) levels, and that opposite changes in β-Klotho expression in fat and the liver indicate potential tissue-specific alterations in the responsiveness to endocrine FGFs in obesity. The associations of TSK with FGF21 and FGF19 need to be investigated in the future.

The circulating level of TSK has been shown to be increased in response to NAFLD and liver damage in humans [5]. The TSK level was shown to be significantly higher in acetaminophen-induced ALF patients, who either died or required emergency transplantation, than in patients who survived without liver transplantation [5]. However, in that study, circulating TSK was under the detection limit in normal control subjects [5], suggesting that the detection sensitivity in the measurement kit is probably lower than that in the present study. It has been reported that the relationship between the gene expression of TSK in the liver and the release of TSK protein in the blood is not linear [5], though the precise identity of the transcription factors that activate the transcription of TSK remains to be determined. It is possible that hepatic stress activates the translation of TSK, decreases the degradation of TSK, or increases the efficiency of TSK secretion. This also needs to be addressed in the future.

The previous study, using mice, showed that TSK reduced circulating HDL cholesterol, lowered cholesterol efflux capacity, and decreased the conversion of cholesterol to bile acid in the liver [5]. However, in the present study, there were no significant correlations between plasma TSK concentration and levels of HDL cholesterol and total bile acids, though the TSK level was independently and negatively correlated with the total cholesterol. Further studies, using a large number of patients with dyslipidemia, are necessary to investigate the association between TSK and cholesterol metabolism, including HDL cholesterol, reverse cholesterol transport, and bile acid synthesis.

Other than liver damage and cholesterol metabolism, the circulating level of TSK was shown to be associated with the risk of hyperthyroidism in multivariable logistic regression analyses, adjusting for age, gender, smoking, BMI, fasting glucose, LDL cholesterol, and insulin resistance [18]. Furthermore, a previous study, using a small number of subjects, showed that the serum TSK level was associated with metabolic disorders in obese subjects (n = 103), but not in non-obese control subjects (n = 41) [19]. In the present study, the TSK level was not associated with adiposity and adiponectin, an adipokine, in a general population of subjects who have not taken any medications (n = 253). It is necessary to show what underlies the associations between TSK and obesity and metabolic disorders in longitudinal studies, using a large number of subjects.

ER stress and inappropriate adaptation through an unfolded protein response are predominant features of pathological processes in several types of tissue [20]. It has been shown that the activation of an unfolded protein response itself induces inflammation, through c-Jun N-terminal kinase-activator protein-1 and κ B kinase–nuclear factor-κ B inhibitor pathways [21,22,23]. Recent studies have revealed relationships between ER stress and a wide range of diseases [20,24,25,26]. However, there have been no useful biomarkers for ER stress [20]. It has been suggested that TSK is produced when the liver faces various stressful conditions linked to excessive lipid flux in the tissue [5,6]. Both ER stress and its related chronic inflammation were reported to be closely linked to excessive lipid deposition in the liver, and to promote expression of TSK and release by the liver [5,6]. Therefore, the circulating level of TSK, mainly derived from hepatocytes, might be a novel biomarker for metabolic-driven ER stress and its related chronic inflammation in the liver.

Several circulating molecules, including adipokines, hepatokines and other factors, are involved in the long-term control of food intake and energy balance [27]. Hepatokines are proteins released by the liver, and they have an impact on various organs to control systemic metabolism [28,29,30]. Growing evidence indicates that changes in several hepatokines can contribute to the development of glucose intolerance, insulin resistance and cardiovascular diseases [28,29,30]. Therefore, hepatokines could represent new targets for the treatment of metabolic and cardiovascular diseases. It is possible that the modulation of TSK can contribute to a novel therapeutic strategy for metabolic and cardiovascular diseases in humans. It is necessary, in the future, to determine whether a change in TSK level, by direct inhibition, neutralization and/or blockade of unidentified receptors, reflects the conditions and outcomes of metabolic and cardiovascular diseases.

There was no significant sex difference in TSK level in the present study (Table 1). However, the correlation coefficients of TSK level with metabolic parameters, including FLI, insulin, HOMA-R and FGF21, were higher in female subjects than in male subjects (Table 2). TSK is also known as E2-induced gene 4 protein (EIG4) or leucine-rich repeat-containing protein 54 (LRRC54), and it has been reported that EIG4 is an estrogen-responsive gene [31]. Sex hormones may affect the associations between TSK and metabolic factors.

This study has some limitations. First, the results obtained in the present study do not prove causal relationships between plasma TSK concentration and the correlated biomarkers, because this study is a cross-sectional study. Second, since only Japanese people were enrolled, the results obtained in the present study might not be applicable to other races. Third, the ELISA kit for TSK, used in the present study, was also used in other studies [18,19], and the specificity of the kit was tested against an array of several cytokines, according to the manufacturer’s protocol. However, more precise validation of the ELISA kit needs to be addressed in the future. Lastly, the change in plasma TSK concentration was not investigated in the present study. Prospective and/or international studies using several drugs for metabolic diseases are needed in the future.

In conclusion, a high plasma TSK concentration is independently associated with high levels of insulin and FGF21, a hepatokine, and a low level of total cholesterol, but not with adiposity and adiponectin, an adipokine, in a general population of subjects who have not taken any medications. A further understanding of the associations between plasma TSK concentration and insulin resistance, hepatic stress and cholesterol metabolism may enable the development of novel therapies for metabolic disease and its related diseases.

## 4. Materials and Methods

### 4.1. Study Subjects

In a population-based cohort, in the Tanno–Sobetsu Study, a total of 627 Japanese subjects (men/women: 292/335) who received annual health examinations in 2016 were recruited. Subjects who were being treated with any medications were excluded to eliminate the effects of drugs on plasma TSK concentration. A total of 253 subjects (men/women: 114/139) were included in the present study. This study was approved by the Ethics Committee of Sapporo Medical University (number: H24-7-30) and was conducted in accordance with the Declaration of Helsinki. Written informed consent was obtained from all of the study subjects.

### 4.2. Measurements

Medical checkups, including measurement of blood pressure and calculation of body mass index (BMI), and collection of blood samples were performed as previously described [32,33,34]. The concentration of TSK was measured using an enzyme-linked immunosorbent assay kit for TSK (RayBiotech, Peachtree Corners, GA, USA). Intra- and inter-assay coefficients of variation were <10% and <12%, respectively. According to the manufacturer’s protocol, no cross-reactivity of TSK with an array of several cytokines occured. Validation was also performed by using a recombinant TSK protein (R&D System, Minneapolis, MN, USA). Concentrations of adiponectin and FGF21 were measured using enzyme-linked immunosorbent assays kits for adiponectin (R&D Systems, Minneapolis, MN, USA) and FGF21 (R&D Systems, Minneapolis, MN, USA), respectively. Variables of liver function, renal function, and glucose and lipid metabolism were measured as previously described [32,33,34]. The level of total bile acids was measured by an enzymatic colorimetric method. Brain natriuretic peptide (BNP) was measured using a commercially available assay kit (Shionogi & Co., Osaka, Japan). Estimated glomerular filtration rate (eGFR) was calculated using an equation for Japanese individuals [35]. As an index of insulin resistance, homeostasis model assessment of insulin resistance (HOMA-R) was calculated as glucose (mg/dL) × insulin (μU/mL)/405 [36]. HbA1c was expressed on the National Glycohemoglobin Standardization Program (NGSP) scale. Fatty liver index (FLI) was calculated, as follows, by using BMI, waist circumference (WC), and levels of γ-glutamyl transpeptidase (GGT) and triglycerides [37]: FLI = [e ^(0.953 × ln (triglycerides) + 0.139 × BMI + 0.718 × ln (GGT) + 0.053 × WC − 15.745)^]/[1 + e ^(0.953 × ln (triglycerides) + 0.139 × BMI + 0.718 × ln (GGT) + 0.053 × WC − 15.745)^] × 100. It has been reported that FLI has high concordance with the histological criteria for NAFLD [37,38,39].

Hypertension was diagnosed as follows, in accordance with the guideline of the Japanese Society of Hypertension [40]: systolic blood pressure ≥140 mmHg or diastolic blood pressure ≥90 mmHg. Diabetes mellitus was diagnosed as follows, in accordance with the guideline of the American Diabetes Association [41]: fasting plasma glucose ≥126 mg/dL or hemoglobin A1c ≥ 6.5%. Dyslipidemia was diagnosed as low-density lipoprotein (LDL) cholesterol ≥ 140 mg/dL, HDL cholesterol < 40 mg/dL or triglycerides ≥ 150 mg/dL, in accordance with the guideline of the Japan Atherosclerosis Society [42].

### 4.3. Statistical Analysis

Numerical parameters are expressed as means ± standard deviations (SD) for normal distributions or medians (interquartile ranges) for skewed variables. The normality of the distribution for each variable was tested by the Shapiro–Wilk W test. Comparisons of parametric and nonparametric parameters between the two groups were performed by using the Student’s *t*-test and Mann–Whitney U test, respectively. The chi-square test was performed for intergroup differences in percentages of parameters. Pearson’s correlation analysis was performed to investigate correlations between two variables. Non-normally distributed variables were logarithmically transformed for regression analyses. Multivariable regression analysis was performed to identify independent determinants of plasma TSK concentration, using age, sex and variables with a significant correlation as independent predictors, after consideration of multicollinearity, showing unstandardized regression coefficient, standard error of regression coefficient and standardized regression coefficient (β), the percentage of variance for the selected independent predictors explained (R^2^), and Akaike’s information criterion (AIC). Parameters with a lower AIC score constitute a better-fit model. A *p* value of less than 0.05 was considered statistically significant. All data were analyzed by using JMP15.2.1 for Macintosh (SAS Institute, Cary, NC, USA).

## Figures and Tables

**Figure 1 metabolites-12-00237-f001:**
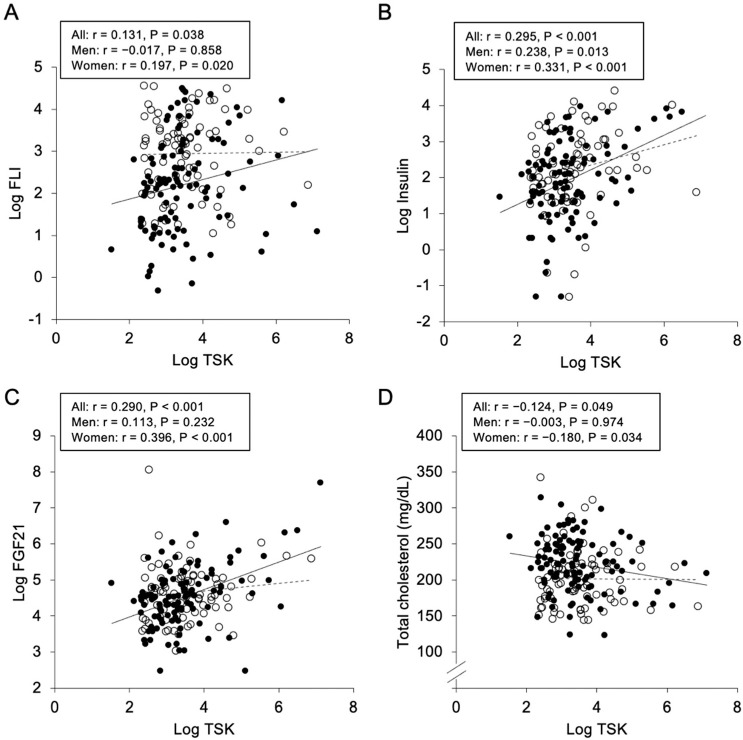
Correlations of Tsukushi concentration with various parameters. Logarithmically transformed (Log) fatty liver index (FLI) (**A**), Log insulin (**B**), Log fibroblast growth factor 21 (FGF21) (**C**) and total cholesterol (**D**) were plotted against Log Tsukushi (TSK) in each subject (n = 253). Open circles and broken regression line: men (n = 114), closed circles and solid regression line: women (n = 139).

**Table 1 metabolites-12-00237-t001:** Characteristics of the studied subjects.

	All (n = 253)	Male (n = 114)	Female (n = 139)	*p*
Age (years)	57 ± 16	56 ± 17	57 ± 16	0.425
Body mass index	22.6 ± 3.8	23.7 ± 3.6	21.8 ± 3.8	**<0.001**
Waist circumference (cm)	82.9 ± 11.5	85.5 ± 10.9	80.7 ± 1.9	**0.001**
Systolic blood pressure (mmHg)	127 ± 20	130 ± 17	124 ± 22	**0.019**
Diastolic blood pressure (mmHg)	74 ± 11	76 ± 10	73 ± 11	**0.024**
Current smoking habit	61 (24.1)	37 (32.5)	24 (17.3)	**0.003**
Alcohol drinking habit	103 (40.7)	61 (53.5)	42 (30.2)	**<0.001**
Comorbidity				
Hypertension	66 (26.1)	29 (25.4)	37 (26.6)	0.832
Diabetes mellitus	3 (1.2)	3 (2.6)	0 (0)	**0.028**
Dyslipidemia	115 (45.5)	46 (40.4)	69 (49.6)	0.139
Biochemical data				
AST (IU/L)	22 (18–26)	23 (20–27)	21 (18–25)	**0.013**
ALT (IU/L)	17 (14–24)	21 (16–29)	16 (13–20)	**<0.001**
GGT (IU/L)	21 (15–32)	26 (20–39)	17 (14–26)	**<0.001**
FLI	13.3 (5.3–32.4)	24.2 (9.2–41.7)	8.8 (3.8–20.4)	**<0.001**
TBA (µmol/L)	4.4 (2.1–8.4)	5.2 (2.6–5.2)	3.8 (2.0–7.3)	**0.042**
Blood urea nitrogen (mg/dL)	15 ± 4	15 ± 4	14.6 ± 4.0	0.122
Creatinine (mg/dL)	0.8 ± 0.2	0.9 ± 0.2	0.7 ± 0.1	**<0.001**
eGFR (mL/min/1.73 m^2^)	73.3 ± 15.2	76.1 ± 15.7	70.9 ± 14.4	**0.007**
Uric acid (mg/dL)	5.2 ± 1.3	6.0 ± 1.1	4.7 ± 1.0	**<0.001**
Total cholesterol (mg/dL)	213 ± 39	200 ± 36	222 ± 38	**<0.001**
LDL cholesterol (mg/dL)	125 ± 34	117 ± 31	132 ± 35	**<0.001**
HDL cholesterol (mg/dL)	64 ± 17	56 ± 16	69 ± 16	**<0.001**
Triglycerides (mg/dL)	79 (59–114)	90 (63–138)	75 (54–107)	**0.003**
Fasting glucose (mg/dL)	89 (85–95)	92 (86–98)	89 (84–93)	**0.003**
Hemoglobin A1c (%)	5.4 ± 0.8	5.5 ± 1.1	5.3 ± 0.4	**0.042**
Insulin (µU/mL)	8.2 (3.8–17.6)	9.1 (4.3–19.3)	7.3 (3.7–15.8)	0.098
HOMA-R	1.78 (0.88–4.02)	1.91 (0.99–4.44)	1.54 (0.84–3.47)	**0.045**
Adiponectin (µg/mL)	7.2 (4.7–10.6)	5.4 (3.7–8.3)	8.9 (6.1–12.5)	**<0.001**
FGF21 (pg/mL)	96 (61–151)	104 (68–158)	92 (55–143)	0.074
BNP (pg/mL)	14 (9–25)	13 (7–24)	16 (11–29)	**0.006**
Tsukushi (ng/mL)	28 (18–49)	33 (19–56)	27 (17–45)	0.086

Variables are expressed as number (%), means ± SD or medians (interquartile ranges). AST, aspartate transaminase; ALT, alanine transaminase; BNP, brain natriuretic peptide; eGFR, estimated glomerular filtration rate; FGF21, fibroblast growth factor 21; FLI, fatty liver index; GGT, γ-glutamyl transpeptidase; HDL, high-density lipoprotein; HOMA-R, homeostasis model assessment of insulin resistance; LDL, low-density lipoprotein; TBA, total bile acids.

**Table 2 metabolites-12-00237-t002:** Correlation analysis for Log Tsukushi (n = 253).

	All (n = 253)		Male (n = 114)		Female (n = 139)	
	r	*p*	r	*p*	r	*p*
Age	−0.047	0.459	−0.086	0.361	−0.009	0.920
Body mass index	0.114	0.069	0.059	0.536	0.128	0.133
Waist circumference	0.108	0.088	0.030	0.750	0.141	0.099
Systolic blood pressure	0.028	0.661	−0.061	0.522	0.062	0.470
Diastolic blood pressure	0.100	0.114	0.032	0.733	0.131	0.127
Biochemical data						
Log AST	0.014	0.821	−0.053	0.578	0.051	0.550
Log ALT	0.089	0.158	0.001	0.992	0.143	0.093
Log GGT	0.067	0.286	−0.089	0.345	0.156	0.067
Log FLI	0.131	**0.038**	−0.017	0.858	0.197	**0.020**
Log TBA	−0.038	0.603	−0.116	0.249	0.031	0.771
Blood urea nitrogen	−0.001	0.991	0.052	0.586	−0.056	0.511
Creatinine	0.119	0.058	0.156	0.098	0.024	0.778
eGFR	−0.032	0.619	−0.116	0.219	0.014	0.873
Uric acid	0.049	0.434	0.060	0.529	−0.026	0.758
Total cholesterol	−0.124	**0.049**	−0.003	0.974	−0.180	**0.034**
LDL cholesterol	−0.107	0.090	0.018	0.852	−0.170	**0.046**
HDL cholesterol	−0.043	0.501	0.135	0.152	−0.125	0.144
Log Triglycerides	0.064	0.309	−0.050	0.598	0.150	0.078
Log Fasting glucose	0.037	0.558	−0.051	0.594	0.133	0.119
Hemoglobin A1c	−0.040	0.524	−0.094	0.322	0.024	0.778
Log Insulin	0.295	**<0.001**	0.238	**0.013**	0.331	**<0.001**
Log HOMA-R	0.293	**<0.001**	0.223	**0.020**	0.338	**<0.001**
Log Adiponectin	0.061	0.333	0.126	0.183	0.069	0.417
Log FGF21	0.290	**<0.001**	0.113	0.232	0.396	**<0.001**
Log BNP	−0.015	0.810	−0.003	0.978	−0.002	0.980

AST, aspartate transaminase; ALT, alanine transaminase; BNP, brain natriuretic peptide; eGFR, estimated glomerular filtration rate; FGF21, fibroblast growth factor 21; FLI, fatty liver index; GGT, γ-glutamyl transpeptidase; HDL, high-density lipoprotein; HOMA-R, homeostasis model assessment of insulin resistance; LDL, low-density lipoprotein; TBA, total bile acids.

**Table 3 metabolites-12-00237-t003:** Multivariable regression analysis for Log Tsukushi.

	Regression Coefficient	SE	Standardized Regression Coefficient (β)	*p*
Age	−0.001	0.003	−0.027	0.671
Sex (Male)	−0.024	0.062	−0.027	0.697
Log FLI	0.033	0.057	0.041	0.561
Total cholesterol	−0.004	0.002	−0.159	**0.020**
Log Insulin	0.229	0.050	0.285	**<0.001**
Log FGF21	0.228	0.076	0.197	**0.003**

R^2^ = 0.154, AIC = 598. AIC, Akaike’s information criterion; FLI, fatty liver index; FGF21, fibroblast growth factor 21.

## Data Availability

The data presented in this study are available on request from the corresponding author. The data are not publicly available due to their containing information that could compromise the privacy of research participants.

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
