# Peer review of "Plasma Tsukushi Concentration Is Associated with High Levels of Insulin and FGF21 and Low Level of Total Cholesterol in a General Population without Medication"

_metabolites, 2022, doi:10.3390/metabo12030237_

Round 1
Reviewer 1 Report
Major.
TSK is emerging as a new hepatokine. Several studies performed in rodents showed that this protein is expressed and secreted by the liver in response to NAFLD and stress. Over the last two years, 2 other groups have investigated the relation between circulating TSK and various clinical parameters in humans. In one paper (Liu et al, 2020; PMID: 33117292), the authors used an ELISA kit from RayBiotech to measure circulating TSK. The kit is the same as the one used here. In the paper of Liu et al., much higher TSK levels were measured (average 100-300ng/ml vs 18-56ng/ml here). Interestingly, another group (Li et al, 2021, PMID : 34329643) used a different ELISA kit from Eiaab Science and measured circulating TSK levels between 0.5ng/ml and 1.74ng/ml (20 to 100x less than the other two studies). Although we expect possible differences between circulating TSK levels in different cohorts, the amplitude of the differences is somehow disturbing.
Overall, I am very concerned by the fact that none of the reports published so far did present results to validate the specificity of the ELISA kits against human TSK. Are the authors sure that the kit they used is specific and only measures TSK? The company (RayBiotech) mentions that the specificity was tested against an array of other cytokines. However, this does not exclude the possibility that their ELISA could recognized other proteins in the plasma that could interfere with the measurement of TSK.
The role of TSK on metabolism is only emerging and there is a real possibility that the field could derive based on the use of a non-specific ELISA kit. Because all the conclusions of this manuscript are derived from the use of one single (and non-validated) ELISA kit, it is imperative for the authors to clearly demonstrate that this kit is specific and only measures TSK. In the case the specificity cannot be demonstrated, I do not see how I could accept the publication of this article unfortunately.
Minor
1-This paper is very well written. The introduction is extremely clear and summarizes the recent discoveries on TSK. The discussion is balanced and interesting. I want to congratulate the authors for their work.
2-It would be interesting to put some signs (*) next to the significant associations in table 2.
Author Response
To Reviewer #1:
Thank you so much for reviewing and providing useful comments on our manuscript entitled “Plasma Tsukushi concentration is associated with high levels of insulin and FGF21 and low level of total cholesterol in a general population without medication” (metabolites-1590388).
The whole manuscript including figures and tables has been revised and rewritten in accordance with your comments.
Our point-by-point responses are presented below.
- TSK is emerging as a new hepatokine. Several studies performed in rodents showed that this protein is expressed and secreted by the liver in response to NAFLD and stress. Over the last two years, 2 other groups have investigated the relation between circulating TSK and various clinical parameters in humans. In one paper (Liu et al, 2020; PMID: 33117292), the authors used an ELISA kit from RayBiotech to measure circulating TSK. The kit is the same as the one used here. In the paper of Liu et al., much higher TSK levels were measured (average 100-300ng/ml vs 18-56ng/ml here). Interestingly, another group (Li et al, 2021, PMID : 34329643) used a different ELISA kit from Eiaab Science and measured circulating TSK levels between 0.5ng/ml and 1.74ng/ml (20 to 100x less than the other two studies). Although we expect possible differences between circulating TSK levels in different cohorts, the amplitude of the differences is somehow disturbing.
In the present study, the median value (interquartile range) of TSK levels measured by using an ELISA kit from RayBiotech was 28 (18-49) ng/mL in a general population without medication (mean age: 57 years, females: 54.9%). In the study by Li et al., the median value of TSK levels measured by using the same ELISA kit was 23.8 (12.5-34.6) ng/mL in control subjects (mean age: 63 years, females: 39.4%), which was similar to the results of the present study. On the other hand, in the study by Liu et al., TSK levels were also measured by using the same kit in patients with hyperthyroidism and those without hyperthyroidism (mean ages: 31 years and 34 years, respectively; females: 69.4% and 59%, respectively), and the median level of TSK was 97.3 (78.9-150.0) ng/mL in the control subjects, which was about 3~4-fold higher than the value in the present study. A possible reason for the different TSK levels is the difference in age of the subjects, though there was no significant correlation between age and TSK level in the present study using middle-aged and elderly subjects. TSK is also known as E2-induced gene 4 protein (EIG4) or leucine-rich repeat-containing protein 54 (LRRC54), and it has been reported that EIG4 is an estrogen-responsive gene (Charpentier AH, et al. Cancer Res 60, 5977-5983, 2000). The relatively high levels of TSK in the study by Liu et al. might be due to the fact that the majority of subjects were premenopausal subjects. We added a limitation about the accuracy of the ELISA kit for detection of TSK in Discussion (p. 9, lines 306-310).
- Overall, I am very concerned by the fact that none of the reports published so far did present results to validate the specificity of the ELISA kits against human TSK. Are the authors sure that the kit they used is specific and only measures TSK? The company (RayBiotech) mentions that the specificity was tested against an array of other cytokines. However, this does not exclude the possibility that their ELISA could recognized other proteins in the plasma that could interfere with the measurement of TSK.
According to the company, RayBiotech, of the ELISA kit that we used in the present study, specificity was tested against an array of several cytokines. However, as you mentioned, it is possible that the ELISA could recognize other proteins in plasma and interfere with the measurement of TSK. Therefore, we accordingly added some sentences about the kit in Methods (pp. 2-3, lines 95-99) and an issue about validation of the ELISA kit as a limitation in Discussion (p. 9, lines 306-310).
- The role of TSK on metabolism is only emerging and there is a real possibility that the field could derive based on the use of a non-specific ELISA kit. Because all the conclusions of this manuscript are derived from the use of one single (and non-validated) ELISA kit, it is imperative for the authors to clearly demonstrate that this kit is specific and only measures TSK. In the case the specificity cannot be demonstrated, I do not see how I could accept the publication of this article unfortunately.
We appreciated your comments. We added this possibility as a limitation in Discussion (p. 9, lines 306-310).
- This paper is very well written. The introduction is extremely clear and summarizes the recent discoveries on TSK. The discussion is balanced and interesting. I want to congratulate the authors for their work.
We appreciated your comments.
- It would be interesting to put some signs (*) next to the significant associations in table 2.
We appreciate your suggestion. According to your suggestion and the suggestion by another reviewer, we used bold letters for p values that are statistically significant in Tables 1-3.
Reviewer 2 Report
Major comments:
This cross-sectional study aimed to investigate associations of plasma TSK concentration with metabolic markers including FGF21 and adiponectin in a general population without medication. The authors should address the following significant issues before further consideration.
1. Introduction: The rationale of excluding medication use to eliminate its effects on plasma TSK concentration should be provided.
2. Methods/Results:
2-1: A quarter of HTN subjects and a half of patients with dyslipidemia did not take medications, which limited the generalizability of this study.
2-2: What was the rationale for defining dyslipidemia with an LDL level ≥ 140 mg/dL?
2-3: The smoking and drinking prevalence was high in this cohort. The authors should validate the representativeness of this cohort, crucial for a cross-sectional study.
2-4: GGT is not always part of routine liver enzymes across settings. The authors should explain why they chose fatty liver index as a non-invasive tool to measure NAFLD in this study.
2-5 (Table 3): Considering multicollinearity, why was insulin a better variable to be incorporated in multivariable regression analyses than HOMA-R? The final model did not include glucose levels. Regression diagnostics should be performed.
3. Discussion: It has been reported that circulating TSK concentrations are independently associated with hyperthyroidism. Some relevant rescue discussions should be added.
Author Response
To Reviewer #2:
Thank you so much for reviewing and providing useful comments on our manuscript entitled “Plasma Tsukushi concentration is associated with high levels of insulin and FGF21 and low level of total cholesterol in a general population without medication” (metabolites-1590388).
The whole manuscript including figures and tables has been revised and rewritten in accordance with your comments.
Our point-by-point responses are presented below.
- Introduction: The rationale of excluding medication use to eliminate its effects on plasma TSK concentration should be provided.
We agree with your comments. We added the rationale for excluding medication use to eliminate its effects on plasma TSK concentration in Introduction (p. 2, lines 78-81).
- Methods/Results:
2-1: A quarter of HTN subjects and a half of patients with dyslipidemia did not take medications, which limited the generalizability of this study.
We appreciate your comments. It has been reported that the prevalence of hypertension in middle-aged and elderly individuals is about 50% in Japan and that about half of the individuals with hypertension in Japan are untreated (Uemura S, et al. Hypertens Res 42: 1235-1481, 2019). On the other hand, the prevalence of dyslipidemia in Japan has been reported to be more than 50% in middle-aged and elderly individuals, and the prevalence is increasing according to the Ministry of Health, Labor and Welfare "National Health and Nutrition Survey" in Japan. Therefore, estimates of about one quarter of subjects with hypertension and about half of the subjects with dyslipidemia would be reasonable in the present study in which middle-aged and elderly subjects without medication.
2-2: What was the rationale for defining dyslipidemia with an LDL level ≥ 140 mg/dL?
According to Japanese guidelines for dyslipidemia (ref. 20), LDL level ≥ 140 mg/dL is defined as dyslipidemia in Japan. We added this information to Methods (p. 3, lines 119-121).
2-3: The smoking and drinking prevalence was high in this cohort. The authors should validate the representativeness of this cohort, crucial for a cross-sectional study.
According to the Ministry of Health, Labor and Welfare "National Health and Nutrition Survey" in Japan (https://www.nibiohn.go.jp/eiken/kenkounippon21/en/eiyouchousa/index.html), the proportions of males and females aged 50-60 years with a smoking habit were about 30% and 15%, respectively, which are similar to those in the present study. On the other hand, alcohol drinking habit was defined without the amount of alcohol consumption in the present study. According to the Ministry of Health, Labor and Welfare "National Health and Nutrition Survey" in Japan, the proportions of males and females aged 50-60 years with an alcohol drinking habit regardless of the amount of alcohol consumption were about 50% and 30%, respectively, which are similar to those in the present study. We accordingly added some sentences to Results (p. 3, lines 143-147).
2-4: GGT is not always part of routine liver enzymes across settings. The authors should explain why they chose fatty liver index as a non-invasive tool to measure NAFLD in this study.
We appreciate your comments. Measurement of GGT is included in routine measurements of liver enzymes in annual health examinations in Japan. FLI calculated by using BMI, waist circumference and levels of triglycerides and GGT has a high concordance with the histological criteria for NAFLD (refs. 15-17). Therefore, calculation of FLI is being used as an alternative method for detection of hepatosteatosis in a general population and in epidemiological studies. Furthermore, abdominal ultrasonography was not a routine examination in the present study. Therefore, we used calculation of FLI as a non-invasive tool to assess NAFLD in the present study.
2-5 (Table 3): Considering multicollinearity, why was insulin a better variable to be incorporated in multivariable regression analyses than HOMA-R? The final model did not include glucose levels. Regression diagnostics should be performed.
Since fasting glucose level was not correlated with TSK, we excluded fasting glucose level in the final model. Considering multicollinearity, we compared insulin and HOMA-R using AIC for choosing a possible candidate of covariates in multivariable regression analyses. Insulin level with a lower AIC score was a better candidate than HOMA-R with a higher AIC score in multivariable regression models. We added this explanation in Methods (p. 3, lines 136-137) and Results (p. 7, lines 194-196).
- Discussion: It has been reported that circulating TSK concentrations are independently associated with hyperthyroidism. Some relevant rescue discussions should be added.
We appreciate your comments. We added the paper showing the association between TSK and hyperthyroidism in Discussion (p. 8, lines 260-263).
Reviewer 3 Report
GENERAL COMMENTS
The manuscript addresses a topic of scientific interest, which is within the journal’s scope. The results presented are orginal and innovative. A broader view of potential underlying mechanisms might be interesting to comment.
The manuscript may benefit from considering the following aspects:
Tables 2 and 3: it would be useful for the reader to highlight in bold letters those p values that are statistically significant.
Discussion:
Metabolic hormones released by endocrine cells and other cell types serve to regulate nutrient intake and energy homeostasis. TSK is a leucine-rich repeat-containing protein secreted primarily by the liver that exerts an inhibitory effect on brown fat sympathetic innervation and thermogenesis. Despite this, physiological regulation of TSK and the mechanisms underlying its effects on energy balance remain poorly understood. It has been shown that hepatic expression and plasma concentrations of TSK are induced by feeding and regulated by melanocortin-4 receptor (MC4R) signaling (ref Wang Q, Zhang P, Cakir I, Mi L, Cone RD, Lin JD. Deletion of the Feeding-Induced Hepatokine TSK Ameliorates the Melanocortin Obesity Syndrome. Diabetes. 2021 Sep;70(9):2081-2091. ). At the cellular level, TSK deficiency augments feeding-induced c-Fos expression in the paraventricular nucleus of the hypothalamus. These results illustrate physiological cross talk between TSK and the central regulatory circuit in maintaining energy balance and metabolic homeostasis.
Serum TSK levels are associated with metabolic disorders in people with obesity. and rs11236956 has been identified in these subjects and with metabolic disorders in the total population (ref Li Y, Jin L, Yan J, Huang Y, Zhang H, Zhang R, Hu C. Tsukushi and TSKU genotype in obesity and related metabolic disorders. J Endocrinol Invest. 2021 Dec;44(12):2645-2654).
It should be mentioned that obesity is characterized by reciprocal alterations in FGF19 (decrease) and FGF21 (increase) levels. Although worsened in diabetic patients, obesity itself appears as the predominant determinant of the abnormalities in FGF21 and FGF19 levels. Opposite changes in β-Klotho expression in fat and liver indicate potential tissue-specific alterations in the responsiveness to endocrine FGFs in obesity (ref Gallego-Escuredo JM, Gómez-Ambrosi J, Catalan V, Domingo P, Giralt M, Frühbeck G, Villarroya F. Opposite alterations in FGF21 and FGF19 levels and disturbed expression of the receptor machinery for endocrine FGFs in obese patients. Int J Obes (Lond). 2015 Jan;39(1):121-9).
It would be also worthwhile mentioning that the involvement of other factors/hepatokines/adipokines can not discarded (ref Frühbeck G, Gómez-Ambrosi J. Rationale for the existence of additional adipostatic hormones. FASEB J. 2001 Sep;15(11):1996-2006.).
Author Response
To Reviewer #3:
Thank you so much for reviewing and providing useful comments on our manuscript entitled “Plasma Tsukushi concentration is associated with high levels of insulin and FGF21 and low level of total cholesterol in a general population without medication” (metabolites-1590388).
The whole manuscript including figures and tables has been revised and rewritten in accordance with your comments.
Our point-by-point responses are presented below.
- Tables 2 and 3: it would be useful for the reader to highlight in bold letters those p values that are statistically significant.
We agree with your comments. We used bold letters for p values that are statistically significant in Tables 1-3.
- Discussion: Metabolic hormones released by endocrine cells and other cell types serve to regulate nutrient intake and energy homeostasis. TSK is a leucine-rich repeat-containing protein secreted primarily by the liver that exerts an inhibitory effect on brown fat sympathetic innervation and thermogenesis. Despite this, physiological regulation of TSK and the mechanisms underlying its effects on energy balance remain poorly understood. It has been shown that hepatic expression and plasma concentrations of TSK are induced by feeding and regulated by melanocortin-4 receptor (MC4R) signaling (ref Wang Q, Zhang P, Cakir I, Mi L, Cone RD, Lin JD. Deletion of the Feeding-Induced Hepatokine TSK Ameliorates the Melanocortin Obesity Syndrome. Diabetes. 2021 Sep;70(9):2081-2091. ). At the cellular level, TSK deficiency augments feeding-induced c-Fos expression in the paraventricular nucleus of the hypothalamus. These results illustrate physiological cross talk between TSK and the central regulatory circuit in maintaining energy balance and metabolic homeostasis.
We appreciate your comments. We cited the paper and mentioned it in Introduction (p. 2, lines 60-65) and Discussion (p. 7, lines 215-217).
- Serum TSK levels are associated with metabolic disorders in people with obesity. and rs11236956 has been identified in these subjects and with metabolic disorders in the total population (ref Li Y, Jin L, Yan J, Huang Y, Zhang H, Zhang R, Hu C. Tsukushi and TSKU genotype in obesity and related metabolic disorders. J Endocrinol Invest. 2021 Dec;44(12):2645-2654).
We appreciate your comments. We cited the paper and mentioned it in Discussion (p. 8, lines 263-269).
- It should be mentioned that obesity is characterized by reciprocal alterations in FGF19 (decrease) and FGF21 (increase) levels. Although worsened in diabetic patients, obesity itself appears as the predominant determinant of the abnormalities in FGF21 and FGF19 levels. Opposite changes in β-Klotho expression in fat and liver indicate potential tissue-specific alterations in the responsiveness to endocrine FGFs in obesity (ref Gallego-Escuredo JM, Gómez-Ambrosi J, Catalan V, Domingo P, Giralt M, Frühbeck G, Villarroya F. Opposite alterations in FGF21 and FGF19 levels and disturbed expression of the receptor machinery for endocrine FGFs in obese patients. Int J Obes (Lond). 2015 Jan;39(1):121-9).
We appreciate your comments. We cited the paper and mentioned it in Discussion (p. 8, lines 235-239).
- It would be also worthwhile mentioning that the involvement of other factors/hepatokines/adipokines can not be discarded (ref Frühbeck G, Gómez-Ambrosi J. Rationale for the existence of additional adipostatic hormones. FASEB J. 2001 Sep;15(11):1996-2006.).
We appreciate your comments. We cited the paper and mentioned it in Discussion (p. 9, lines 285-286).
Reviewer 4 Report
I read with interest the manuscript by Furuhashi and colleagues on the hepatokine Tsukushi levels in the general population.
TSK is involved in several developmental processes. It is released mainly by the liver and seems to regulate gluco-lipidic pathways. Recently, Li YY et al (Diabetes Res Clin Pract 2021) have shown that TSK is higher in diabetic patients and other authors stated that it is induced under oxidative stress conditions in the context of NAFLD.
In this study the authors show that there is a significant correlation between plasma levels of TSK and FGF21, another hepatokine which is involved in the pathogenesis of NAFLD. Moreover, TSK levels are associated with insulin concentration. These data suggest the potential involvement of TSK in the onset of insulin resistance in the setting of NAFLD even if in the study cohort, the prevalence of type 2 diabetes was low and insulin and glucose levels were in the normal range (including HOMA index below 2.7).
More studies are needed to completely understand the role of TSK in the context of NAFLD as well as in other clinical conditions such as obesity, metabolic syndrome or diabetes. Notwithstanding this, these data suggest that TSK could be investigated as pathogenetic factor involved in the onset and progression of NAFLD.
Minor comments:
- The best correlations between TSK and other metabolic factors are in the female group. The authors should thoroughly discuss this difference.
- In the Methods section "Measurements", the authors should better describe the ELISA kit characteristics including the CV of the kit
- In the discussion section, the authors should improve the paragraph describing molecular pathways involving TSK and FGF21
- I suggest to perform the analysis splitting the study cohort according to the presence of metabolyc syndrome (considering MetS if present 3/5 figures among glucose levels, waist circumference, HDL cholesterol levels, triglycerides levels and blood pressure/hypertension) to make the manuscript more appealing from a clinical point of view.
Author Response
To Reviewer #4:
Thank you so much for reviewing and providing useful comments on our manuscript entitled “Plasma Tsukushi concentration is associated with high levels of insulin and FGF21 and low level of total cholesterol in a general population without medication” (metabolites-1590388).
The whole manuscript including figures and tables has been revised and rewritten in accordance with your comments.
Our point-by-point responses are presented below.
- The best correlations between TSK and other metabolic factors are in the female group. The authors should thoroughly discuss this difference.
We agree with your comments. We do not know the exact mechanisms of the better correlations of TSK with other metabolic markers in female subjects. However, TSK is also known as E2-induced gene 4 protein (EIG4) or leucine-rich repeat-containing protein 54 (LRRC54), and it has been reported that EIG4 is an estrogen-responsive gene (ref. 42). Sex hormones may affect the associations of TSK with other metabolic factors. We added this possibility in Discussion (p. 9, lines 296-301).
- In the Methods section "Measurements", the authors should better describe the ELISA kit characteristics including the CV of the kit.
We agree with your comments. Intra-and inter-assay coefficients of variation were <10% and <12%, respectively. We added this information in Methods (p. 3, lines 97-98).
- In the discussion section, the authors should improve the paragraph describing molecular pathways involving TSK and FGF21.
We agree with your comments. According to your suggestion and the suggestion by another reviewer, we amended the paragraph describing molecular pathways involving TSK and FGF21 (p. 8, lines 228-239).
- I suggest to perform the analysis splitting the study cohort according to the presence of metabolic syndrome (considering MetS if present 3/5 figures among glucose levels, waist circumference, HDL cholesterol levels, triglycerides levels and blood pressure/hypertension) to make the manuscript more appealing from a clinical point of view.
We agree with your comments. According to your suggestion, we investigated the relationship of TSK with metabolic syndrome (MetS). There was no significant difference in TSK level between subjects with MetS (n = 36) and those without MetS (n = 217) (35.8 [24.5-60.8] vs. 27.7 [17-47.6] ng/mL, P = 0.065).
Round 2
Reviewer 1 Report
The main concern I [still] have about this paper is that the kit the author used to measure TSK levels was not validated. The authors rely only on the statement of the company that sells this kit (there is obviously a conflict of interest here). Unfortunately, the authors made no attempt to validate this tool. Because all the conclusions of this manuscript are derived from the use of this non-validated ELISA kit, it is imperative for the authors to clearly demonstrate that the kit is specific and only measures TSK. In the case the specificity cannot be demonstrated, I do not see how I could accept the publication of this article unfortunately.
The authors mention in their response to my comment : 'However, as you mentioned, it is possible that the ELISA could recognize other proteins in plasma and interfere with the measurement of TSK'. This means all their conclusions might be wrong. This is major. This paper should not be published unless the authors convincingly show their ELISA is specific to measure TSk.
Author Response
- The main concern I [still] have about this paper is that the kit the author used to measure TSK levels was not validated. The authors rely only on the statement of the company that sells this kit (there is obviously a conflict of interest here). Unfortunately, the authors made no attempt to validate this tool. Because all the conclusions of this manuscript are derived from the use of this non-validated ELISA kit, it is imperative for the authors to clearly demonstrate that the kit is specific and only measures TSK. In the case the specificity cannot be demonstrated, I do not see how I could accept the publication of this article unfortunately.
We performed analyses of standard curve using a commercially available recombinant TSK protein (R&D Systems) and preset recombinant TSK protein in the RayBiotech ELISA kit. We confirmed high linearity of diluted samples (R2 ≥ 0.99) using both recombinant proteins and comparable concentration. Furthermore, in addition to the validation by RayBiotech using several cytokines including human C1qTNF9, Carbonic Anhydrase VA, CANT1, Cathepsin H, Contactin-5, Chymotrypsin C/CTRC, Draxin, EphB2, EphB3, FABP8, Fgr, FKBP51, FUCA1, Galanin, GALNT10, Gastrokine 1, Glyoxalase II, HS3ST1, HS3ST3B1, Lin28, Lysyl Oxidase Homolog, LRRTM4, Methionine Aminopeptidase 1D, Matrilin-2, MCEMP1, Mcl-1, MDGA2, MEF2C, Methionine Aminopeptidase 2, Neurocan, Nogo-A, PCK1, PlGF-2, PON1, SALM4/LRFN3, Semaphorin 6C, SorCS2, ST3GAL1, ST8SIA1, we also confirmed no cross-reactivity with FGF21 (hepatokine), adiponectin (adipokine) and FABP4 (adipokine). At least, TSK can be measured by the kit which we used, though we cannot totally deny that the ELISA could recognize other proteins in plasma. We according added this information in Methods (p. 3, lines 100-101).
- The authors mention in their response to my comment: 'However, as you mentioned, it is possible that the ELISA could recognize other proteins in plasma and interfere with the measurement of TSK'. This means all their conclusions might be wrong. This is major. This paper should not be published unless the authors convincingly show their ELISA is specific to measure TSk.
We appreciate your comments. We deleted the sentence.
Reviewer 2 Report
The background of medication effects on plasma TSK concentration is still not provided. I suggest the medications at least should cover common anti-hypertensive and lipid-lowering drugs.
Author Response
- The background of medication effects on plasma TSK concentration is still not provided. I suggest the medications at least should cover common anti-hypertensive and lipid-lowering drugs.
We agree with your comments. We added the background of medication effects on plasma TSK concentration (p. 2, lines 78-83).